# Acceptability and feasibility of using contingency management for cannabis reduction in specialist mental health services for psychosis: A qualitative study of staff views

**Laura Middleton Curran**[1,2,3]**, Luke Sheridan Rains**[1]***, Jo Taylor**[1,4]**, Nicola Morant**[1]**, Sonia Johnson**[1,2]

**1** Division of Psychiatry, University College London, London, United Kingdom, **2** Camden and Islington NHS Foundation Trust, London, United Kingdom, **3** University of East London, London, United Kingdom, **4** Public Health England, London, United Kingdom

* l.sheridanrains@ucl.ac.uk

## Abstract

### Aim

There is increasing evidence linking cannabis use to onset, continuation, and relapse of psychosis. Contingency Management (CM) is discussed as a candidate intervention to reduce cannabis use. Our study aimed to explore staff views on the feasibility and acceptability of using CM for cannabis reduction in early intervention services for psychosis (EIS), in order to inform wider learning about implementation of such approaches in mental health services.

### Setting

EIS teams in England.

### Method

Semi-structured interviews and focus groups analysed thematically.

### Participants

Forty managers and staff members working in mental health services where a CM intervention was delivered as part of a trial, four staff who delivered CM in these settings, and three key informants (academic experts in relevant fields).

### Intervention

A complex intervention comprising CM with incremental financial incentives (vouchers) for reducing or stopping cannabis use, and psychoeducation about the risks of cannabis use.

**Data Availability Statement:** The data relevant to this study are restricted as they contain sensitive and potentially identifying information. Requests

for access should be sent to Prof. Bryn Lloyd-Evans, University College London, at b.lloyd-evans@ucl.ac.uk or to the Mental Health Policy Research Unit (MHPRU) at circle.trial.ucl@gmail.com.

**Funding:** The study was conducted in the context of the CIRCLE trial. The Circle Study was funded by the NIHR HTA programme (ref 09/144/50). The views expressed are those of the authors and not necessarily those of the NHS, the NIHR or the Department of Health. The funders had no role in study design, data collection and analysis, decision to publish, or preparation of the manuscript.

**Competing interests:** The authors have declared that no competing interests exist.

## Findings

Acceptability appeared to depend on how well the intervention was seen to fit with the service setting and ethos. Concerns included who should deliver CM; potential impacts on the therapeutic relationship; the ethics of using incentives to reduce socially objectionable behaviours; and how CM fits with the work of mental health practitioners. Feasibility concerns centred on resource limitations including time, cost, training, and national guidance and commissioning.

## Conclusions

Staff attitudes are likely to be a crucial influence on successful implementation of contingency management for cannabis reduction in specialist mental health settings. Several contextual barriers would need to be overcome to increase the acceptability of the intervention for use in early intervention services for psychosis.

## 1 Introduction

Contingency management (CM) is an intervention for behaviour change that involves reinforcing behaviours such as reduction or abstinence from substance use, often with monetary rewards. There is a substantial and growing evidence base demonstrating its efficacy for a range of substance misuse problems [1, 2] and cohorts, including individuals with co-morbid mental health difficulties [3]. In one systematic review of CM, Davis et al. [4] reviewed 69 studies of CM for illicit substance use, alcohol, and tobacco. Of these, 86% found a positive effects of CM compared to controls and comparators at the end of treatment. Of 28 studies that measured longer-term outcomes, only eight found some evidence of continued benefits after discontinuation of CM, and the authors outline the need for future research to focus on sustaining longer-term outcomes [4]. A Cochrane review of psychosocial interventions for comorbid substance misuse and mental health diagnoses included two randomised controlled trials (RCTs) of contingency management, although neither were for cannabis cessation. The authors found no significant effect of CM from these studies but concluded that the limited available evidence for this population was low quality [5, 6].

Despite some promising initial results and subsequent clinical recommendations from the National Institute of Clinical Excellence (NICE) [7, 8] and Department of Health and Social Care [9], CM has not been widely adopted by NHS services in England. Questionnaire data from a study of the views of addiction counsellors in New Hampshire indicate that the use of CM may also have been limited in the USA, and that it is still rated as the least acceptable psychosocial intervention compared with others such as cognitive behavioural therapy (CBT) [10]. Acceptability has been identified as a significant potential barrier to international implementation of CM: various qualitative research data from focus groups and interviews highlights staff apprehensions that CM may be coercive [11–14], paternalistic [11, 14, 15], and potentially damaging to therapeutic relationships [12, 16]. Additional concerns have centred on how individuals might spend rewards, particularly cash [12–16], and on CM being perceived a reward for 'bad' behaviour [13, 14]. Furthermore, staff voiced uncertainty about the public perception of use of public funds for CM [14]; the opportunity cost of CM versus other interventions [14]; and the risk that CM could be viewed as a cheap alternative to other provisions, resulting in funding cuts [12]. Acceptability is a multi-faceted concept [17], which is

vital to consider in the development and evaluation of complex interventions, as it influences the likelihood of an intervention being adopted, successfully delivered, and sustained by services [18].

This qualitative study is the first of its kind to explore staff's views of the acceptability and feasibility of using CM for cannabis reduction in the context of mental health services for young adults experiencing psychosis. Psychosis has a typical onset in adolescence and young adulthood and young people are particularly susceptible to cannabis use (e.g. prevalence of use of over 8% in the UK [19]). Supporting those who use cannabis and attend Early Intervention in Psychosis Services (EIS) to reduce or stop using cannabis if they want to, may help with reducing relapse of symptoms for these young people. This study was conducted in the context of the CIRCLE trial, which was an RCT investigating the clinical and cost-effectiveness of CM for cannabis use in EIS teams [20]. The trial did not provide clear evidence supporting the clinical effectiveness of CM in this population at the level of reward offered. Secondary analyses suggested that the intervention may have been clinically and cost-effective among those who attended CM sessions, but that participant disengagement was an important potential contributor to failure to attain clear effectiveness [21]. CIRCLE was novel for being one of the largest trials of CM in a clinical population, and the first of CM for treating cannabis use in people with psychosis. It was delivered in 70 EIS across 23 NHS trusts (healthcare regions) in England. EIS are multidisciplinary, specialist, community-based mental health services that provide intensive support to people within three years of their first treated episode of psychosis [22]. Given the size of the CIRCLE trial, and that it is one of the few instances of CM being delivered in mental health services in England, it offers an excellent opportunity to explore the views of managers and staff in the mental health services regarding the acceptability and feasibility of CM for substance use cessation within specialist mental health services. Findings can potentially inform decisions about whether CM should be further applied and tested in this population, how its implementation could be maximised, and implementation strategies for CM more widely.

### 1.1 Aims and objectives

The study aims to explore staff views on the acceptability and feasibility of using a CM plus psychoeducation intervention to reduce cannabis use in the context of specialist mental health services for young adults with psychosis.

## 2 Methods

### 2.1 Setting

The CIRCLE trial was a pragmatic, multi-centre RCT conducted in EIS teams in the Midlands, East, and South East of England, including London [20, 21]. Participation in the trial by EIS teams was voluntary and followed a process of approach and onboarding by the research team, which included (amongst other stages) the opportunity to discuss the aims, procedures, and resource requirements of the trial and its interventions, receiving necessary organisational approvals, and the provision of equipment and training in trial procedures and intervention delivery. Participants were young adults (18–36 years) within the first 3 years of treatment for psychosis who regularly used cannabis. Eligible service users were identified by healthcare workers within participating EIS teams, and consented and assessed by the research team. Participants at each EIS team were randomized at individual level to either the experimental or control arm of the study [21]. Participants in the experimental arm received a 12-week CM program with an optimised treatment-as-usual psychoeducation package, while those in the control group received the psychoeducation package only. The CM involved 12 once-weekly

urinalysis sessions, delivered face-to-face by a mental health worker, and were delivered either in the EIS site itself, in participants' homes, or in another suitable location. The interventions were delivered by clinical staff within the EIS teams, including care coordinators, healthcare assistants, and those who were in pre-qualified roles in allied health professions, such as assistant psychologists. This varied between sites depending on resources and capacity within teams and was agreed with the research team. Suitable procedures and training were developed for healthcare workers delivering the CM and psychoeducation interventions such that they could have been used if the interventions were implemented in routine EIS practice. Participants received shopping vouchers if the results indicated reduced or no cannabis use in the previous week; the total possible voucher reward was £240 –the voucher schedule was variable over the course of the trial with the value of vouchers increasing with subsequent urine samples negative for cannabis (for full details see: Sheridan-Rains et al., 2019). The psychoeducation package comprised six, once-weekly sessions based on motivational interviewing, relapse prevention, and harm minimisation principles. It included information about the potential role of cannabis in increasing risk for psychosis onset and relapse [20]. For participants in the experimental group, the psychoeducation and CM were typically delivered together for 6 weeks, and CM alone for 6 weeks. Further details of the CIRCLE trial are presented in Johnson et al. [20] and Sheridan Rains et al. [21].

## 2.2 Participants

Purposive sampling was used to select and recruit participants from three stakeholder groups: EIS managers; multidisciplinary EIS staff working in trial sites but *not* delivering the CM intervention; and staff who directly delivered CM. Our sampling aims were to include staff from a range of different roles, professions, levels of experience, ages, genders, and ethnic groups, working in services in rural and urban areas, and to include core members of the EIS, such as care coordinators, in order to explore acceptability and feasibility of the intervention. To contextualise this within current thinking on addictions research and management, a small number of key informants who were academic experts on CM interventions were also interviewed.

## 2.3 Data collection

Individuals who directly delivered CM, EIS service managers, and key informants, took part in semi-structured interviews. Focus groups were conducted with EIS team staff, none of whom had delivered CM personally, but worked in teams where the intervention was delivered. In both interviews and focus groups participants shared their views of: acceptability of CM; feasibility of implementing CM in EIS; perceived benefits of and barriers to CM; and ethics, policy, and implementation of CM. Schedules for semi-structured interviews and focus groups were developed by the research team based on the interview materials and results of a similar qualitative project conducted following the pilot phase of the CIRCLE trial [23], as well as on previous literature on acceptability of CM interventions [11–13, 16]. Three interviews were conducted by telephone due to participant preference, all other focus groups and interviews were held at participants' places of work. Data was collected by LMC and JT, both jointly and independently. Written consent was obtained before data collection, participants received no payment. Focus groups and interviews lasted around 30 minutes on average (range: 14 to 45 minutes), were audio-recorded, and transcribed verbatim.

## 2.4 Data analysis

Thematic analysis was used to systematically code and explore data using NVivo (version 11). The analysis process followed guidance by Braun and Clarke [24] incorporating stages of:

familiarisation; generation of initial codes; review and revision of the data; and exploration of patterns and relationships [24]. LMC led on the analysis with contributions from NM and SJ. LMC and LSR began by familiarising themselves with the data. Initial codes were constructed inductively from the data by LMC and reviewed by LSR. Through regular meetings between researchers, these codes were refined and organised into a thematic framework. The thematic framework was refined iteratively through multiple meetings to enhance reflexivity and analytic validity by discussing meanings and emerging themes, and sharing perspectives on the data. Concept maps were used to visually organise data into codes and themes. The final thematic framework was approved by the authors.

## 2.5 Authors

At the time of the analysis, four of the five authors worked as university-based researchers in the field of mental health, including one who is a senior psychiatrist with substantial experience of working in EIS and who has researched cannabis use in early psychosis over many years. One was an early-career mental health professional. All authors were White British, four were female and one male.

## 2.6 Ethics approval

Ethical approval for the present study was received as part of substantial amendment no. 6 to CIRCLE on 22nd November 2016 by the London–South East NRES committee (REC reference 11/LO/1939) [20].

# 3 Results

We completed interviews with six EIS team managers (one had delivered CM directly), four EIS team members who had delivered CM, and three key informants who were academic experts who have published on a variety of aspects of delivery and effectiveness of CM interventions. Five focus groups were conducted with a total of 36 EIS staff members (Table 1), none of whom had personally delivered the CM intervention as part of the CIRCLE trial, though they all worked in trial sites. Participants came from a range of professional backgrounds that are typically part of EIS staff teams in the UK, including psychiatrists, nurses, occupational therapists, and social workers (see Table 2). All participants worked in EIS teams in urban and suburban areas of London.

Three main themes were identified: the first, EIS staff's concerns around the ethics of using CM for cannabis reduction, was balanced against their enthusiasm for new evidence-based interventions in this area. The second theme was the (in)compatibility of CM with current working practices and clinical approaches within EIS, including concerns about how aspects of CM may erode therapeutic relationships. The third theme centred on resource considerations—time, cost, and wider systemic processes, and competing priorities relating to clinical guidelines and commissioning.

**Table 1. Data collection.**

| Stakeholder Group | Data Collection Method | Participants n = 47 |
|---|---|---|
| **EIS Managers**[*] | Interviews | 6 |
| **Individuals who delivered CM** | Interviews | 4 |
| **EIS Staff** | Focus Groups (n = 5) | 36 |
| **Academic Key Informants** | Interviews | 3 |

**Table 2. EIS staff demographics.**

| | | N (total = 44) (%) |
|---|---|---|
| Gender | Female | 29 (66%) |
| | Not known | 2 (5%) |
| Age | 20–29 | 8 (18%) |
| | 30–39 | 8 (18%) |
| | 40–49 | 12 (27%) |
| | 50–59 | 6 (14%) |
| | 60+ | 1 (2%) |
| | Not known | 9 (20%) |
| Ethnicity | White British | 20 (45%) |
| | White Other | 7 (16%) |
| | Black British | 2 (5%) |
| | Black African | 3 (7%) |
| | British Asian | 2 (5%) |
| | Asian Other | 6 (14%) |
| | Irish | 1 (2%) |
| | Not Known | 3 (7%) |
| Years of experience | ≤5 | 13 (30%) |
| | 6–10 | 4 (9%) |
| | 11–15 | 4 (9%) |
| | 16–20 | 10 (23%) |
| | 21–25 | 3 (7%) |
| | >25 | 2 (5%) |
| | Not known | 8 (18%) |
| Job Roles | Team Manager | 5 (11%) |
| | Psychiatrist | 6 (14%) |
| | Care Coordinator | 5 (11%) |
| | Nurse | 5 (11%) |
| | Student Nurse | 2 (5%) |
| | Clinical Psychologist | 2 (5%) |
| | Assistant Psychologist | 3 (7%) |
| | Social Worker | 4 (9%) |
| | Social Work Student | 1 (2%) |
| | Support Worker | 3 (7%) |
| | Occupational Therapist | 3 (7%) |
| | Research Assistant | 2 (5%) |
| | Family Intervention Practitioner | 1 (2%) |
| | Not Known | 2 (5%) |

*Two managers were also present at their teams' focus groups

## 3.1 EIS staff views of contingency management for cannabis use reduction: Ethics and evidence

Overall, managers and staff supported testing whether CM helped their clients reduce their cannabis use. Cannabis use was recognised almost universally by staff as a problem for many EIS service users. There was a widespread view that cannabis, especially if high potency, has negative effects in psychosis. CM was seen as a potentially acceptable

intervention, proportionate to these difficulties. Managers, individuals who delivered CM, and EIS staff in the focus groups tended toward a pragmatic view of CM best summarised as: if it works, use it. Evidence was viewed as the best means of persuading any doubting staff to support it.

Staff views were congruent with those of expert key informants, who argued that using CM to support cannabis cessation is ethical provided the individual consents to the intervention and is personally motivated to reduce their cannabis use. One key informant suggested it could be considered unethical *not* to use CM if proven effective.

*". . . you're. . .trying to help somebody overcome a problem, and you are using one treatment and you have got that in one pocket and in your other pocket you have got an enhancer to the treatment. . . what possible reason have you got for not maximising the benefit?"*

(Key informant 1 interview)

Some staff also saw potential benefits of CM in increasing overall engagement with services, seen as a general barrier to delivering effective interventions in EIS. In this context, CM was viewed as a viable 'last resort' where other strategies had not worked. There were however mixed views on whether CM would in practice improve engagement. Some felt that financial incentives might encourage attendance, whilst others thought this would not be effective for the most disengaged. Managers were in general keen to try novel interventions such as CM where other alternatives have not been effective.

*". . .whatever interventions we've tried in the past. . .haven't really been that effective. So, I was all in favour of trying something which was new and. . .innovative. . .to get people to reduce their cannabis use."*

(Manager 5 interview)

Some ethical concerns were raised by participants. Some managers and staff from the focus groups (who had not delivered the intervention) worried about seeming paternalistic or coercive, and expressed concerns about the erosion of trust that 'policing' cannabis use via urine testing might provoke. Other ethical issues included discomfort with monetary incentives, which were perceived to compromise personal agency and function as 'bribes' for individuals to behave in particular ways. One staff member admitted they would not want to use CM in their personal practice even if there were good evidence of effect:

*"I'm accountable for my own practice . . . I personally wouldn't feel very comfortable with it, getting into an area of work where I'm doing [urine drug screens] with someone, giving them money. It's a very long way from the way that I've worked with people. . . I'd be interested in the results of the study because it's useful. . . new evidence, but I think resistance, for myself, would be a barrier. I wouldn't be very keen on doing it."*

(Staff Member 1 focus group 5)

A second ethical issue that several staff raised was that some clients may use monetary rewards to buy harmful or illegal substances. As such, staff tended to perceive vouchers as less risky than cash incentives. However, most participants who voiced concerns balanced this by suggesting this could be over-ridden by evidence of effectiveness:

*"I guess if it works . . . if the data shows that it has a big impact, then, you know, and it's cost effective, then why not? So, I think we just feel uneasy about it involving money."*

(staff member 2, focus group 2)

*"I remember people being, like, oh, make sure you say they can't spend it on drugs and alcohol. . . or cigarettes was the main one"*

(Individual who Delivered CM 3 interview)

### 3.2 Service fit: Therapeutic relationships and training

**3.2.1 Therapeutic relationships and delivery of CM.**   Many staff wondered how compatible CM would be with the current ethos and working practices of EIS. Debate centred on who should deliver CM. The majority of EIS staff were concerned about the possible negative impacts of urine-testing and payment in CM on therapeutic relationships and trust. Few staff supported the idea of integrating CM into care coordinators' roles if evidenced to be effective. Several said that a reasonable method for integrating CM into EIS care would be for someone external to the core mental health team, such as assistant psychologists or placement staff, to deliver it to their clients, as largely occurred in the CIRCLE trial.

*"[some staff] have a slight objection to kind of urine testing people. So, they think that, you know, you should go with what the client tells us. So, if the client tells you they're not using [cannabis] that's what you should work with . . . taking urine samples from someone . . . suggests that you don't believe them. And, so, that can be a bit of a barrier."*

(Manager interview 5)

Those who had delivered CM were generally more positive about impacts on therapeutic relationships than those who had not. Some described having had concerns about therapeutic relationship impacts before delivering the intervention, but reported largely positive experiences with CM in practice. They described CM as collaborative, feeling pleased to be able to offer something tangible to service users who made an effort to engage:

*". . . it can be collaborative, it can be interesting, it can be a shared project, it can be a shared learning experience. . . looking at the different results, what do they mean? You know, it can involve. . . praise, encouragement, and support, it can involve,. . . thinking about how when something doesn't go the way you wanted it to. . . how do you pick up from that. . . I think there's a lot of stuff you can do. . . around those aspects."*

(Individual who Delivered CM 1 interview)

Key informants' views on therapeutic relationship were more complex. One discussed the potential for incentives to threaten trust, but noted also that informal incentives are used within mental health services already: thus, what makes staff uncomfortable may be making the use of incentives explicit. Another suggested staff may find the idea that a financial incentive could be more effective than therapy difficult, due to a *"myth of being great fixers and being fantastic helpers"*. This view was echoed in one of the staff member focus groups.

*"it's a bit sad to think that giving someone £10 is more effective than spending an hour with. . . them talking. . . Just from a purely, sort of, narcissistic therapist point of view."*

(Staff Member 3, focus group 3)

**3.2.2 Training.** Staff who had not delivered the CM intervention felt they would need training in both CM and in understanding and providing psychoeducation about the evidence linking cannabis use with psychosis, seeing this as a barrier to its delivery. Training on CM elements that challenged existing working practices and ethos, such as urine testing and use of monetary rewards was considered a possible facilitator to using CM, and lack of training in these areas was considered a potential barrier to future delivery. Additionally, some staff noted that many of the young people they support are highly educated and present their own views of cannabis use. Staff expressed a need for training to build their confidence in conveying messages about cannabis cessation, supported by increasing evidence of the causal link between recreational cannabis use and psychosis for some service users.

*". . .there's also conflicting research and opinions on cannabis and its effect on psychosis. I don't necessarily feel comfortable. . . I need some research to back up why I'm telling people to stop. . . what statistics do I have, or am I just. . .following a political agenda rather than. . .an evidence based one?. . .a lot of the young people we work with are very bright. They're at university, studying PhDs. They're more educated than we are. . .we really do need good evidence to back up what we're saying. . ."*

(Staff Member 3, focus group 1)

### 3.3 Resources

Scarcity of NHS resources was raised as a barrier to implementing CM in EIS in every interview and focus group with managers and staff. Three main resource barriers emerged: limited practitioner time, direct costs of implementing CM, and competing priorities resulting from clinical guidelines and commissioning strategies. High caseloads resulting in limited time availability of staff was the most referenced resource barrier:

*"If it's done properly, it would take. . .quite a bit of time and finding that time would,. . .be quite difficult within the current constraints."*

(Staff Member 4, focus group 5)

Concerns about the costs of delivering CM often focused on the initial outlay of purchasing vouchers. However, arguments about costs were also bound up with more complex or ambivalent views about the ethics of CM, resource distribution, and short-term versus long-term costs.

*"I obviously understand that in the long term this is saving a lot of money, if it works. . . having the money right there and handing it over. . . feels more real than just giving them an hour of someone's time. . . it feels more like you're just throwing money away,. . . even though that time. . . costs more money, and then obviously the outcomes in the long run would save so much more money."*

(Individual who Delivered CM 2 interview)

Some staff members worried that CM could divert money away from individuals who do not use cannabis, or who have stopped. This was perceived as unfair and even as a potential risk to recovery for previous users.

*"I was thinking, like, how would other service users perceive it as well? So, maybe people who haven't been using cannabis for a while or haven't even thought about using it and they're seeing other people in the service getting rewarded for stopping this behaviour. . .Oh, maybe I should start this so that I can get some vouchers."*

(Staff Member 5, focus group 3)

In contrast, all key informants centred their arguments around health economic evidence. They saw CM as cost-effective when it worked and tended to focus more on opportunity cost and outcomes.

Ultimately though, many managers and staff thought that resource availability to deliver CM would depend on external factors, particularly how evidence is translated into clinical guidance. Recommendations of CM use in national clinical guidelines and specific or mandated funds for CM from local commissioners of EIS were cited as instrumental requirements for feasible implementation. However, one key informant highlighted that the uptake of CM interventions in the addiction field has been poor despite inclusion in national guidelines [7, 9].

*"I think it's been hugely disappointing, how slow the treatment field has been to, even to recognise the evidence presented to it, uh, and incredibly slow to incorporate this ability to get more benefit from the treatments they are providing. . . some services do provide it, but remarkably slow when the guidance was pretty clear."*

*(Key informant 1 interview)*

## 4 Discussion

This paper explores the views of EIS staff, managers, and other stakeholders on the acceptability and feasibility of using CM to reduce cannabis use within specialist mental health services for young people with psychosis in England. Key findings centred around ethics and evidence, fit with service ethos and approach, and resources suggest only modest practitioner acceptability in its current form, and have implications for CM use in mental health services more widely. The conceptualisation of acceptability proposed by Sekhon et al. [17] as comprising seven constructs: affective attitude, burden, perceived effectiveness, ethicality, intervention coherence, opportunity costs, and self-efficacy offers a useful framework for understanding these staff views of a novel intervention, and the factors that may facilitate or limit the feasibility of its implementation in mental health settings.

CM is sparsely used in NHS services for treating substance misuse, despite being recommended in clinical guidelines [7, 9]. It is not clear why this is the case, but ethical and moral concerns may remain, despite emerging evidence of efficacy. In the current study, CM was viewed by some EIS staff as potentially coercive, with vouchers seen as a 'bribe' rather than a reward or incentive, echoing previous findings on views of CM [11–16, 25–27]. Some staff were concerned about appearing paternalistic and 'policing' or making moral judgements about service users' drug use. They wanted to preserve the autonomy of individuals, a theme also found in previous research [13, 15, 27]. At the same time, staff supported the use of vouchers rather than cash to reduce the risk of incentives being 'misused' by being spent on harmful

substances, an apprehension about CM that is common [12, 14–16]. Ethical concerns were most common amongst those EIS staff who took part in the focus groups, who had not themselves delivered CM. Among staff who had delivered it, views of CM were more positive. Some who had had concerns prior to delivering CM came to view it more positively subsequently, as a way of engaging clients in care, or that could have a positive effect on therapeutic relationships. They had experienced delivering CM as a collaborative process, and were pleased to offer something tangible to their clients for achieving their goals.

In terms of participants' 'affective attitudes' [17], all managers and most staff participants were positive about the use of CM if research evidence supported its effect, thus meeting the acceptability criteria of 'perceived effectiveness'. Despite this, it was a challenge to recruit core team members, such as care coordinators, to take part in the delivery CM as part of the intervention arm of the CIRCLE RCT. This related to the perceived 'burden' of the added workload, as well 'ethicality', as staff voiced concerns about the possibility that CM would disrupt therapeutic relationships and take more time. As a result, the intervention was delivered in many cases by staff other than the service user's care coordinator and other staff responsible for their care. There was ambivalence about how well CM fits with the culture of EIS services, with many viewing CM as too different to the current ethos of these teams to make it feasible in these settings.

Perceived opportunity costs also shaped participants' views of the acceptability and feasibility of CM in EIS context. They expressed concerns about current funding scarcity in the NHS, the direct outlay of monetary incentives, and that funding for CM would unfairly divert resource from EIS service users not using drugs. Most managers felt CM could be a useful adjunct to psychological treatments if there were clear evidence of cost-effectiveness. There was a majority view across staff and managers of the need for a strong mandate in the form of national guidance accompanied by specific commissioning to implement any new interventions, such as CM, into services. However, some key informants and staff suggested that reluctance might stem less from concerns about costs than aversion to the idea that a monetary incentive might prove more effective than the professional therapeutic relationship.

Overall, most managers and staff expressed views in keeping with those of the key informants–that CM is ethical provided there is informed consent. All managers and most of the staff expressed a willingness to accept CM as an intervention, provided there was evidence of efficacy. That said, there was a reluctance from staff to deliver the intervention, and some concerns around ethicality common to previous research on CM. This broadly supports the key informants' views that even with growing evidence of effect, clinicians remain reluctant to use CM in practice.

## 4.1 Strengths and limitations

None of the EIS staff in the focus groups had delivered the CIRCLE CM intervention themselves, but all worked in services delivering CM and many worked with clients who received CM; although several reported disappointment that their clients were in the control group so did not receive the rewards. Their views about CM may have changed had they delivered the intervention, as was reported by other staff who had done so, but are nonetheless valuable for understanding the perceived acceptability and potential barriers to CM delivery within mental health services; particularly as there was reluctance for delivering the intervention, and a preference for others, e.g. assistant psychologists, to do so. Secondly, the intervention was delivered in the context of the RCT, so staff views and experiences may differ from those of an intervention delivered in a more naturalistic setting. There may have been a more structured and supported context for implementation of the CIRCLE intervention than is typical in routine

practice, as well as additional barriers to staff seeing this as core clinical work with a pre-existing evidence base.

Participants represented a range of ages, ethnicities, job roles, seniority and service contexts. The CIRCLE trial recruited EIS teams across England serving urban and rural populations, however in the current study it was only feasible to recruit staff in London. There are some key differences in resources and workload between urban and rural EIS teams which may limit the generalisability of these findings to rural areas. In particular, services in inner London tend to have higher caseloads, so may find it more difficult to implement novel interventions. Recruitment of key informants was purposive—we deliberately selected professionals with specialist knowledge of policy, ethics, and implementation of CM.

## 4.2 Implications

This research provides insights into staff views regarding the acceptability and feasibility of CM for reducing cannabis use in EIS services. These are crucial to the successful implementation and sustainability of complex interventions. Views centred on how well the CM intervention fits with the service ethos and the resources available. There were some ethical concerns regarding CM, such as concerns about 'bribing' or coercing their clients, and reservations about potentially negative effect on therapeutic relationships, which were most often expressed by people with no experience of delivering the intervention. But these were asserted to be secondary to a preference for evidence-based interventions. Presenting evidence of the efficacy of CM, and highlighting guidance recommending its use, is likely to improve staff receptiveness and confidence, especially with their concern about service users with high education levels who question the evidence in this area. Further, presenting the experiences of staff who have delivered the intervention could help clinicians come to a view regarding the potential positive and negative impacts of CM on their therapeutic relationships. Resources were considered to a substantial barrier to implementation in mental health settings, with the burden of delivering CM being considered too high without a clear mandate and additional funding from commissioners. It may be beneficial to introduce CM as an adjunct to existing psychosocial interventions to decrease perceived burden and enhance intervention coherence. Staff raised concerns around feeling 'deskilled' by the novel CM approach. Addressing staff self-efficacy through training on the theory and practicalities of CM is another way in which acceptability may be increased. Results can inform researchers, service managers, commissioners, and policymakers on the possible barriers, and facilitating factors to implementing CM within mental health services in the UK and elsewhere.

## Acknowledgments

The authors would like to thank the EIS managers, staff, and the key informants for agreeing to take part in the interviews and focus groups, and the CIRCLE trial team for their continued support.

## Author Contributions

**Conceptualization:** Laura Middleton Curran, Luke Sheridan Rains, Jo Taylor, Nicola Morant, Sonia Johnson.

**Data curation:** Laura Middleton Curran, Luke Sheridan Rains, Jo Taylor.

**Formal analysis:** Laura Middleton Curran, Luke Sheridan Rains, Nicola Morant.

**Funding acquisition:** Sonia Johnson.

**Methodology:** Laura Middleton Curran, Luke Sheridan Rains, Nicola Morant, Sonia Johnson.

**Supervision:** Luke Sheridan Rains, Nicola Morant, Sonia Johnson.

**Writing – original draft:** Laura Middleton Curran.

**Writing – review & editing:** Laura Middleton Curran, Luke Sheridan Rains, Jo Taylor, Nicola Morant, Sonia Johnson.

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
