## [Decision Letter · Decision Letter 0]

8 Jun 2022

PONE-D-21-36088Acceptability and feasibility of using contingency management for cannabis reduction in specialist mental health services for psychosis: A qualitative study of staff viewsPLOS ONE

Dear Dr. Sheridan Rains,

Thank you for submitting your manuscript to PLOS ONE. After careful consideration, we feel that it has merit but does not fully meet PLOS ONE’s publication criteria as it currently stands. Therefore, we invite you to submit a revised version of the manuscript that addresses the points raised during the review process.

Thank you for your submission, the manuscript is well written and reports results with relevant findings for health professionals and consumers.   The study topic fits within the scope of the journal however some revisions are requested for clarification of your findings.  In particular,  you note that many participants in the focus groups did not have first hand experience with the intervention - were the findings from these participants any different from those who did have first hand experience?

Please consider the reviewers feedback below, in particular the request for more details on the participants. 

We look forward to receiving your revised manuscript.

Kind regards,

Kathleen Finlayson

Academic Editor

PLOS ONE

Journal Requirements:

The study was conducted in the context of the CIRCLE trial. The Circle Study was funded by the NIHR HTA programme (ref 09/144/50). The views expressed are those of the authors and not necessarily those of the NHS, the NIHR or the Department of Health.

The Circle Study was funded by the NIHR HTA programme (ref 09/144/50). The views expressed are those of the authors and not necessarily those of the NHS, the NIHR or the Department of Health.

However, funding information should not appear in the Acknowledgments section or other areas of your manuscript. We will only publish funding information present in the Funding Statement section of the online submission form. 

The study was conducted in the context of the CIRCLE trial. The Circle Study was funded by the NIHR HTA programme (ref 09/144/50). The views expressed are those of the authors and not necessarily those of the NHS, the NIHR or the Department of Health.

None

Reviewers' comments:

Reviewer's Responses to Questions

**Comments to the Author**

1. Is the manuscript technically sound, and do the data support the conclusions?

Reviewer #1: Partly

2. Has the statistical analysis been performed appropriately and rigorously? 

Reviewer #1: N/A

3. Have the authors made all data underlying the findings in their manuscript fully available?

Reviewer #1: No

4. Is the manuscript presented in an intelligible fashion and written in standard English?

Reviewer #1: Yes

5. Review Comments to the Author

Reviewer #1: Thank you for this interesting and well written article.

The main area for clarification and improvement in the paper relates to the inclusion of participants who had and hadn’t used CM, and how that affected the results. You do acknowledge this in your limitations, but I think the paper could be improved so it can be better understood how much this has limited your findings.

-There was not enough information about the CIRCLE trial to allow understanding of the staff participants in this research. You refer to all EIS staff being from trial sites but need to clearly state whether the intervention was delivered in all trial sites. Did people at trial sites have a choice not to implement CM, or did it depend on their role or just the randomisation process? Who actually delivered the intervention within the CIRCLE trial? From your methods it appears it was mental health workers but in the discussion you refer to care coordinators.

- The term focus group is often misused and debated. It seems inaccurate to call your group interviews focus groups given participants within the groups differed in such a fundamental way related to the intervention you are exploring which you then make one of your major points in your discussion. They are really group interviews based on workplace. That said, I am happy for you to continue to call them focus groups given this is a descriptive qualitative study and many such researchers call this focus groups, however cannot resist the comment.

-It is not clear how many staff in the FGs had delivered the intervention and which voices were heard within the FG findings, yet the findings are largely merged into the results and discussion as a whole. In general when you refer to staff in this paper it is done in a way that suggests most staff were against CM and management were for it, as if this was a finding in your results. Yet the FGs included staff with experience and staff without. At one point in the results you specifically note where some data pertained to participants without personal experience of CM but otherwise it is not evident. This becomes a major discussion point that there was a difference between those who experienced CM and those who didn’t, which is interesting but which does not come out well in your results.

Additionally I have some minor suggestions for improvement as below.

Intro Page 3

The introduction set the scene well.

“Of 28 studies that measured longer-term outcomes, eight found some evidence of continued benefits after discontinuation of CM, although other studies have not found evidence of continuing benefits (4,5).” This sentence seemed vague in the contest of the rest of your paragraph. Do you meant to say 20 of 28 showed no evidence of long term effects. It would seem important to expand.

Methods

Setting – please add in the monetary value of the shopping vouchers here

I find the description of a urinalysis intervention unclear – was this a short moment at the end of the education session where the urinalysis was followed by the voucher for participants randomised to the study arm?

Results

The section 1 first quote seemed unnecessary, there are a lot of descriptive quites about evidence already.

6. PLOS authors have the option to publish the peer review history of their article (what does this mean?). If published, this will include your full peer review and any attached files.

Reviewer #1: No

---

## [Author Response · Author response to Decision Letter 0]

20 Jan 2023

Reviewer #1: Thank you for this interesting and well written article.

• The main area for clarification and improvement in the paper relates to the inclusion of participants who had and hadn’t used CM, and how that affected the results. You do acknowledge this in your limitations, but I think the paper could be improved so it can be better understood how much this has limited your findings.

Author Response:

As noted below, we have now clarified how many participants had delivered the intervention throughout the paper.

• There was not enough information about the CIRCLE trial to allow understanding of the staff participants in this research. You refer to all EIS staff being from trial sites but need to clearly state whether the intervention was delivered in all trial sites. Did people at trial sites have a choice not to implement CM, or did it depend on their role or just the randomisation process? Who actually delivered the intervention within the CIRCLE trial? From your methods it appears it was mental health workers but in the discussion you refer to care coordinators.

Author Response:

We agree that this will help improve our manuscript. We have added more information about the CIRCLE to the Setting section in our Methods. 

• The term focus group is often misused and debated. It seems inaccurate to call your group interviews focus groups given participants within the groups differed in such a fundamental way related to the intervention you are exploring which you then make one of your major points in your discussion. They are really group interviews based on workplace. That said, I am happy for you to continue to call them focus groups given this is a descriptive qualitative study and many such researchers call this focus groups, however cannot resist the comment.

Author Response:

Thank you and your reflections are welcome. 

• It is not clear how many staff in the FGs had delivered the intervention and which voices were heard within the FG findings, yet the findings are largely merged into the results and discussion as a whole. In general when you refer to staff in this paper it is done in a way that suggests most staff were against CM and management were for it, as if this was a finding in your results. Yet the FGs included staff with experience and staff without. At one point in the results you specifically note where some data pertained to participants without personal experience of CM but otherwise it is not evident. This becomes a major discussion point that there was a difference between those who experienced CM and those who didn’t, which is interesting but which does not come out well in your results.

Author Response:

This is a valid point. We have clarified how many participants had delivered the intervention throughout the paper, including in the Data Collection section of our Methods, in the first paragraph of our Results section, second paragraph of the Discussion, and also in our Strengths and Limitations section. We have also clarified the source of our findings throughout our Results section.

• Additionally I have some minor suggestions for improvement as below.

Intro Page 3

The introduction set the scene well.

“Of 28 studies that measured longer-term outcomes, eight found some evidence of continued benefits after discontinuation of CM, although other studies have not found evidence of continuing benefits (4,5).” This sentence seemed vague in the contest of the rest of your paragraph. Do you meant to say 20 of 28 showed no evidence of long term effects. It would seem important to expand.

Methods

Setting – please add in the monetary value of the shopping vouchers here

I find the description of a urinalysis intervention unclear – was this a short moment at the end of the education session where the urinalysis was followed by the voucher for participants randomised to the study arm?

Results

The section 1 first quote seemed unnecessary, there are a lot of descriptive quites about evidence already.

Author Response:

We thank you for these additional improvements and we have followed your suggestions.

---

## [Editor Report · Decision Letter 1]

23 Jan 2023

Acceptability and feasibility of using contingency management for cannabis reduction in specialist mental health services for psychosis: A qualitative study of staff views

PONE-D-21-36088R1

Dear Dr. Sheridan Rains,

We’re pleased to inform you that your manuscript has been judged scientifically suitable for publication and will be formally accepted for publication once it meets all outstanding technical requirements.

Kind regards,

Kathleen Finlayson

Academic Editor

PLOS ONE

Additional Editor Comments (optional):

Thank you for your careful revision and response to the reviewer comments, the article provides some valuable information on this intervention. As a final check, could you please ensure all aspects of the appropriate EQUATOR guidelines have been covered (i.e. COREQ or SRQR guidelines), I can see most are well explained, except perhaps for inclusion of your methodological/theoretical framework guiding your methods.

---

## [Editor Report · Acceptance letter]

30 Jan 2023

PONE-D-21-36088R1 

Acceptability and feasibility of using contingency management for cannabis reduction in specialist mental health services for psychosis: A qualitative study of staff views 

Dear Dr. Sheridan Rains:

I'm pleased to inform you that your manuscript has been deemed suitable for publication in PLOS ONE. Congratulations! Your manuscript is now with our production department. 

Kind regards, 

on behalf of

Dr. Kathleen Finlayson 

Academic Editor

PLOS ONE